# An Innovative Treatment Approach Using Digital Workflow and CAD-CAM Part 2: The Restoration of Molar Incisor Hypomineralization in Children

**DOI:** 10.3390/ijerph17051499

**Published:** 2020-02-26

**Authors:** Esti Davidovich, Shlomi Dagon, Israel Tamari, Michael Etinger, Eitan Mijiritsky

**Affiliations:** 1Department of Pediatric Dentistry, Hebrew University—Hadassah School of Dental Medicine, Jerusalem 91120, Israel; 2Advanced Esthetic Program, Center of Aesthetic Dentistry and Clinical Research, Prosthodontic Department, Faculty of Dentistry, Tel Aviv University, Tel-Aviv 6139001, Israel; dsdagonclinic@gmail.com; 3Private Clinic, Tel-Aviv 6139001, Israel; ie.tamary@gmail.com; 4Department of Prosthodontics, Hebrew University—Hadassah School of Dental Medicine, Jerusalem 91120, Israel; dr.etinger@gmail.com; 5Department of Otolaryngology Head and Neck Surgery and Maxillofacial Surgery Tel-Aviv Sourasky Medical Center, Sackler School of Medicine, Tel Aviv University, Tel-Aviv 6139001, Israel; mijiritsky@bezeqint.net

**Keywords:** CAD-CAM, molar incisor hypomineralization, children

## Abstract

Until recently, the treatment for molar incisor hypomineralization (MIH) mainly included interim restorations such as resin restorations and stainless-steel crowns. These require replacement after adolescence. The use of intraoral scanners (IOS) has opened a new venue for restoring MIH teeth, by reducing the challenge of dealing with uncooperative children’s behavior and enabling tooth structure preservation and long-lasting restoration. We present an innovative treatment approach for children with MIH, using a digital workflow with IOS and CAD-CAM (computer-aided design and computer-aided manufacturing) fabrication of the restoration. The overall protocol involves a thorough diagnostic phase throughout treatment planning, which takes into consideration the child’s behavior and the parent’s cooperation and compliance. Initial preparation consists of inhalation sedation if needed, an effective local anesthesia, and the use of a rubber dam. Removal of all areas of enamel and dentin porosity is essential, and the tooth/teeth must be appropriately prepared to accommodate inlays or onlays for molars and labial veneers for incisors. IOS impressions are taken, including scanning of the prepared tooth and its antagonist, scanning of the bite, and CAD-CAM preparation of the restoration. Next is restoration, cementation, and follow up. Digital workflow provides definitive restorations in young patients due to the high accuracy of the scanning.

## 1. Introduction

Molar incisor hypomineralization (MIH) was first described in 2001 by Weerheijm et al. [1], as hypomineralization of systemic origin of one to four permanent first molars, commonly with the involvement of incisors [1]. The range of prevalence reported globally for MIH is wide, 3%−44% [2], as is the variability in clinical presentation. This corresponds with the lack of standardization of a research protocol and the differences that have been observed between samples of children [2]. The documentation of multiple affected molars in four of five children who were deemed to have MIH supports the possibility that some children are more susceptible than others [3].

The cause of MIH remains elusive. Several factors were proposed in the literature: respiratory tract problems [4,5], oxygen starvation, low birth weight, calcium and phosphate metabolic disturbances, frequent childhood illnesses [6], and environmental conditions that affect natal and early development. The latter includes conditions common in the first 3 years of life, such as asthma and upper respiratory diseases, otitis media, tonsillitis, chicken pox, measles, and rubella. The use of antibiotics has also been found to be associated; however, it is difficult to discern whether the association with MIH is due to the antibiotic drug or the pathological condition for which the antibiotic is administered [1,6]. The pathogenesis of disrupted ameloblastic function during late secretory and early maturation stages of amelogenesis is probably the cause of MIH [6,7,8].

Poor general and systemic health more often presents in persons with developmental defects of enamel. Examples of such conditions are nutritional deficiencies, brain injury and neurologic defects, cystic fibrosis, syndromes of epilepsy and dementia, nephrotic syndrome, atopia, repaired cleft lip and palate, radiation treatment, rubella embryopathy, epidermolysis bullosa, ophthalmic conditions, celiac disease, and gastrointestinal disorders [6,9,10].

Restoring teeth with MIH is extremely difficult in children, due to several factors: the challenge in attaining anesthesia, sensitivity and rapid development of dental caries, repeated marginal breakdown of restorations, and the challenge of obtaining the patient’s cooperation.

Intraoral scanners (IOS) are important devices that have become an integral component of the dental tool arsenal. They are used for optical impressions and can access information on the size and shape of dental arches [11,12]. Accordingly, a beam of light grid (structured light or laser) is projected onto the tooth surface, and by means of high-resolution cameras, the reflection of the beam or grid on these structures is captured [13]. The cameras integrate the data by means of software that recreates the three-dimensional (3D) model of the anticipated structures [12,13]. Specifically, a polygonal mesh that represents the scanned object is generated from the genesis of a "cloud of points". The scan is then processed to attain the final 3D model [11]. The conservative physical detection, with trays and materials (alginates, silicones, polyethers) of an impression, is very inconvenient for patients [14,15,16]. IOS enable taking impressions very quickly, and no materials and trays are needed. This is particularly advantageous for children and for persons with a strong gag reflex [17]. Moreover, IOS were shown to be as accurate as their conventional analogues for single teeth or short span bridges [17,18].

Therefore, IOS facilitate taking impressions in children easily, quickly, and accurately. Moreover, optical impressions with IOS can resolve difficulties for clinicians that arise with conventional impression detection and especially with technically complex impressions [13,14,17]. Therefore, IOS are well tolerated by children, because they do not require the use of conventional impression materials, are easy to perform, and are faster for the dentist and for the child [16,17,18].

In this article, we present an innovative treatment approach for children with MIH, using a digital workflow with IOS and CAD-CAM (computer-aided design and computer-aided manufacturing) fabrication of the restoration. The protocol will be illustrated by a clinical case. The treatment approach presented here can facilitate and improve the treatment of children with MIH.

## 2. Methods

The suggested steps of the procedure protocol of this treatment approach are described in Figure 1.

The overall protocol involves a thorough diagnostic phase.

Diagnostic phase: careful examination and documentation of medical and dental history. Attention should be given to pregnancy, birth, and the first year of life. Information on systemic illness should be obtained. It is crucial to diagnose the number of molars affected and the severity of the enamel porosity in each tooth.

The treatment plan should consider factors that are firstly related to the child and his family. The child’s behavior, parent’s cooperation, and compliance are important factors. Furthermore, if there is more than one tooth affected, time scheduling should be considered.

Initial preparation: nitrous oxide inhalation sedation should be considered in children if there is lack of cooperation, and an effective local anesthesia must be administered to the child. Rubber dam placement is crucial for the preparation. The removal of all areas of enamel and dentin porosity is essential, and the tooth/teeth must be appropriately prepared to accommodate overlays for molars and labial veneers for incisors.

Digital workflow: this should consist of IOS impressions, including scanning of the prepared tooth and its antagonist, scanning of the bite, and CAD-CAM preparation of the restoration.

Definitive restoration: restoration bonding is a very sensitive stage. To gain the child’s cooperation, nitrous oxide, an effective local anesthesia, and rubber dams are essential.

Follow-up: the child should be invited for a checkup one week after cementation, three months later, and on a regular basis according to the risk assessment for caries (Figure 1).

### 2.1. Compliance with Ethical Standards

This study received no external funding. All the authors declare no conflicts of interest with this work. All the procedures were in accordance with the ethical standards of the institutional and/or national research committee and with the 1964 Helsinki Declaration and its later amendments or comparable ethical standards.

### 2.2. Clinical Case

The following treatment plan for overlay–veneer restoration in a nine-year-old child illustrates the treatment protocol.

An eight-year-old girl who was caries-free had a hypoplastic #36 molar. The tooth was very sensitive, and the child could barely drink or eat cold and hot beverages and food. Moreover, the tooth was previously restored with amalgam restoration that failed (Figure 2). The parents were seeking a definitive restoration and were opposed to full coverage with a stainless-steel crown due to aesthetic reasons. The lower left first molar had areas of yellow-brownish demarcated opacities of the enamel and areas of extensive post eruptive breakdown. Behavior management of the child was very difficult, and nitrous oxide as inhalation sedation was used.

First, the girl was administered inhalation sedation due to her fear of the procedure. She expressed excitement regarding the scanner and the 3D model. However, she was less enthusiastic regarding the treatment itself (i.e., the local anesthesia, preparation, and cementation). After administering local anesthesia, the upper and the lower jaws and the bite occlusion were scanned using intraoral scanner Primescan connect® (Dentsply Sirona Dental Systems GmbH Bensheim, Germany) software connect version no. 5.1.0 (Dentsply Sirona Dental Systems GmbH Bensheim, Germany) (Figure 2 and Figure 3).

A rubber dam was placed, and the tooth was prepared to receive an overlay restoration. The preparation should include removal of porosities combined with shape preparation. Immediate dentin sealing was administered due to the high sensitivity of MIH teeth while waiting for final restoration and for improving adhesion. The prepared tooth was scanned after removing the rubber dam. Telio® (Ivoclar Vivadent, Schaan, Liechtenstein) was selected as the material for temporary restoration. For the final biomechanical preparation of the tooth, diamond burs with fine abrasiveness were used to eliminate porosities of the tooth.

The girl was scheduled one week later for cementation of the prepared Lithium disilicate (LS2) glass-ceramic (IPS e.max Lithium Disilicate®, Ivoclar Vivadent, Schaan, Liechtenstein) CAD-CAM restoration (Figure 4). Local anesthesia and a rubber dam were used. Variolink® (Ivoclar Vivadent, Schaan Liechtenstein) was selected as the bonding agent (Figure 5). One week later, at the follow-up appointment, the restoration was inspected clinically for integrity of marginal fit and occlusion.

## 3. Discussion

The use of overlays and restorations in the pediatric population is not common. This is because lack of cooperation makes conventional impression taking very challenging in children and frequently even impossible. Moreover, the usual treatment for MIH mainly entails interim restorations such as resin restorations and stainless-steel crowns (SCC). The choice of these treatments must consider replacement of the restorations after adolescence. The introduction to the prosthetic field of restorative digitalization protocols, with IOS and CAD-CAM, has opened a new venue for pediatric dentists and practitioners who treat young children. Such approaches can help deliver the best definitive restorative treatment to MIH teeth in this population. To the best of our knowledge, the treatment approach presented here for children with MIH was not previously described in the literature.

Table 1 presents restorative options for managing MIH. The table describes the decision-making considerations facing pediatric dentists dealing with the restoration of teeth of children with MIH.

We will address the advantages and disadvantages of each technique and demonstrate why the suggested treatment protocol is preferable for MIH in our view, after case selection.

### 3.1. Treatment Options

The choice of materials depends on the severity of the defect and the age and cooperation of the child [19]. Restorative options include the following: resin composites (RC), resin-modified glass ionomer cements (RMGIC), glass ionomer cements (GIC), stainless-steel crowns, and indirect adhesive overlays or crowns.

#### 3.1.1. Adhesive Restorations—RC and Glass Ionomer Cements

Following removal of hypomineralized enamel, atypical cavity outlines remain; thus, adhesive materials are usually selected [20]. Despite the sparse literature regarding adhesion to hypomineralized enamel, the evidence suggests the possibility of 5% sodium hypochlorite pretreatment of the enamel for removal of the protein encasing the hydroxyapatite. The removal of all hypomineralized enamel before RC restoration has been recommended [21,22,23]. The literature is unclear regarding bonding to hypomineralized enamel; this is probably due to a lack of extracted hypomineralized teeth with suitable surfaces for bond strength testing.

Bond strengths are significantly less for RC to MIH teeth than for normal enamel. This is true for all types of bonding materials available (single-bottle total etch and self-etching primer adhesives) [24]. Williams et al. [24] reported significantly lower mean microshear bond strengths of RC bonded to hypomineralized enamel than for control enamel. Scanning electron microscopy revealed interprismatic spaces of hypomineralized enamel after phosphoric acid etching and very little intercrystal porosity within the enamel prisms. This enables limited microtag formation and weaknesses, possibly resulting in crack propagation within the enamel [24].

RC are recommended for restoring molar surfaces with limited involvement. Following the removal of all discolored hypomineralized enamel, cavity margins should be placed on presumably sound enamel, and RC should be bonded with a self-etching primer adhesive. The location of marginal placement on sound enamel is important because of the low adhesion of RC to hypomineralized enamel [24].

The main disadvantages of RC are the following: shrinkage due to the extent of the restoration, reduced strength due to impaired bond strength, microleakage, occlusal wear, and restoration durability.

#### 3.1.2. Adhesive Restorations—Glass Ionomer Cements

GIC confer a few advantages: (1) fluoride release, (2) easy placement, and (3) chemical bonding [20,25,26]. GIC are recommended for dentin replacement or as interim restorations. The benefits of RMGIC are comparable to those of GIC. These include better integration of resin and photo-initiators: (1) wear resistance; (2) fracture toughness; (3) handling; and (4) fracture resistance [27]. Stress-bearing areas, such as occlusal surfaces of hypomineralized molars, are not recommended to be restored with GIC and RMGIC. However, such restoration may be adequate temporarily, until a definitive restoration can be attained [20,25,26,27]. The RC have superior physical properties compared to GIC and RMGIC. In addition, they provide an aesthetic solution, with high wear resistance and adhesion when used with resin-based adhesives. These materials may be applied alone or in a sandwich technique after temporization with GIC.

#### 3.1.3. Full Coronal Coverage Restorations—Stainless-Steel Crowns

SSC are the mainstay therapy for teeth with moderate to severe hypoplasia [6,23,28]. The rationales for full coverage restoration are the following: preventing additional tooth deterioration, control of tooth sensitivity, and the formation of proper interproximal contacts and correct occlusal relationships. SCC are not as technique-sensitive and are less expensive than cast restorations; they also need minimal time to prepare and insert [20,28,29]. If not placed properly, however, SSC may create an open bite or lead to gingivitis [15]. Properly adapted, SSC can preserve teeth with MIH until cast restorations are possible [15,29,30].

#### 3.1.4. Porcelain Veneers and Overlays

Partial and full coverage indirect adhesive crowns and overlays should be well thought out for MIH in late mixed and permanent dentitions [20,28,29,30]. Such restorations are seldom used in young children due to complications in placement associated with the following: (1) short crowns; (2) large pulps; (3) prolonged treatment; (4) high expense; and (5) children’s difficulty in cooperating [15]. However, among children of primary school age, laboratory-fabricated crowns of cast gold, indirect composite, and ceramics demonstrated effectiveness in the five years after treatment [15].

Overlay restorations necessitate less tooth reduction compared to SSC, minimize pulpal trauma, provide high strength for cuspal overlays, protect tooth structure, control sensitivity, and preserve periodontal health, owing to supragingival margins [15,31]. Some have claimed that there are no differences in quality or longevity between indirect adhesive restorations and preformed SSC [29]. In choosing between indirect adhesive restorations and preformed SSC for the restoration of hypomineralized molars, the following are among the factors to be considered:The child’s cooperation;The child’s immediate and long-term needs;Financial costs;The dentist’s skills and the materials available.

## 4. Conclusions

Until recently, the treatment for MIH included mainly interim restorations such as resin restorations and SCC. These restorations require replacement after adolescence. The use of IOS has opened a new venue for restoring MIH teeth by reducing the challenge of children’s behavior and enabling tooth structure preservation and long-lasting restoration. Lately, digital dentistry has become more accessible and evidenced-based in daily dental practice. According to the authors’ experience, digital workflow should be one of the preferred choices for the treatment of MIH, because it provides definitive restorations in children due to the high accuracy of the scanning. Children’s behavior, family preference, and the severity of the teeth affected remain the main factors to be considered.

## Figures and Tables

**Figure 1 ijerph-17-01499-f001:**
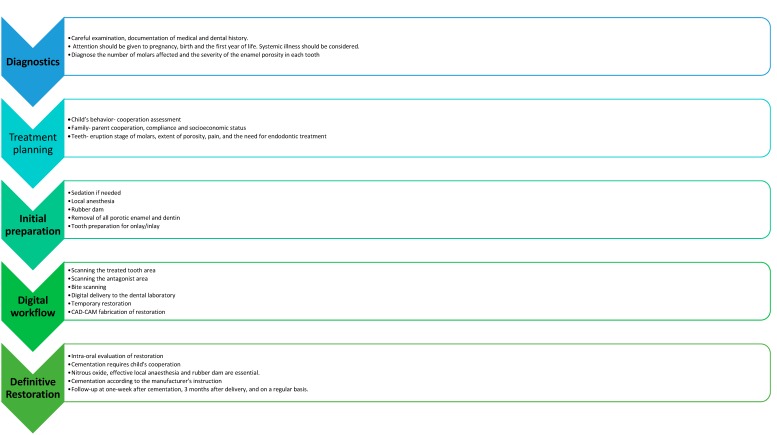
The suggested steps for the protocol of treatment approach in molar incisor hypomineralization molars in children.

**Figure 2 ijerph-17-01499-f002:**
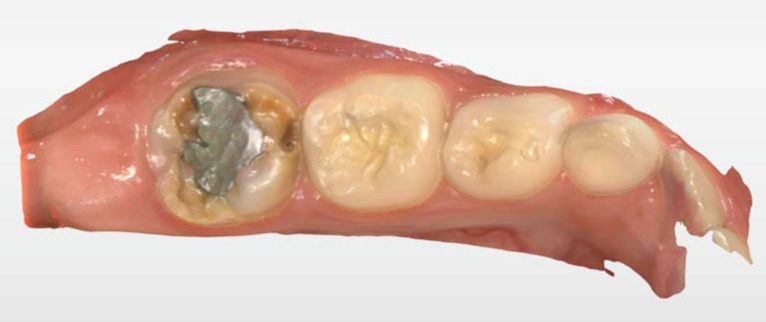
A scan of tooth #36 before tooth preparation.

**Figure 3 ijerph-17-01499-f003:**
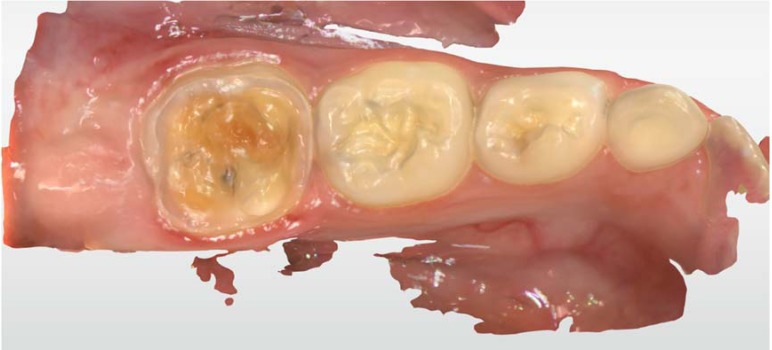
A scan of tooth #16 after tooth preparation.

**Figure 4 ijerph-17-01499-f004:**
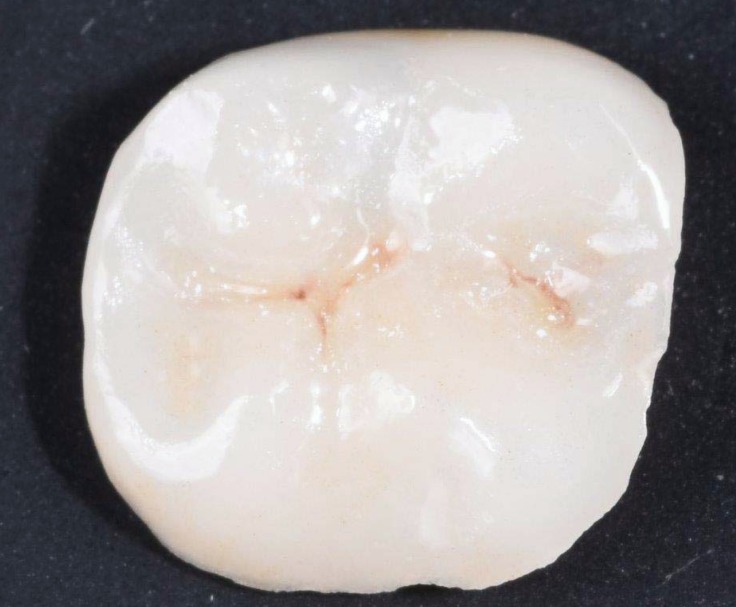
The prepared overlay restoration before cementation.

**Figure 5 ijerph-17-01499-f005:**
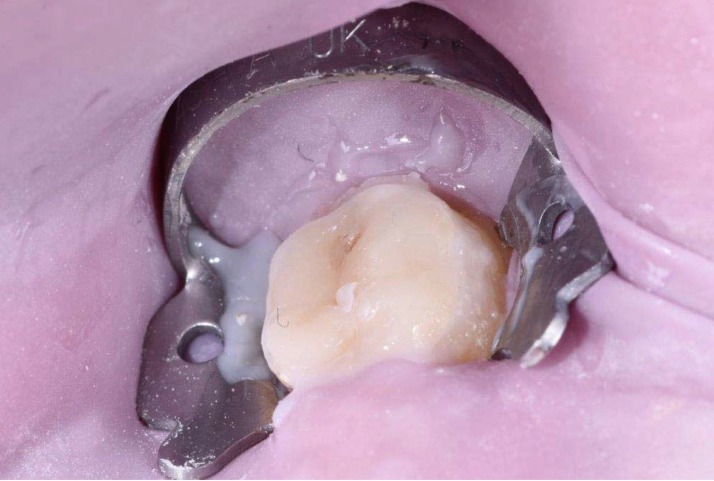
Cementation of restoration.

**Table 1 ijerph-17-01499-t001:** Decision-making considerations when dealing with the restorative options for molar incisor hypomineralization (MIH) teeth in children.

Restorative Considerations	Composite Restoration	Glass Ionomer Restoration	Full Coverage Stainless-Steel Crown	CAD/CAMCeramic Overlays
Effectiveness of restorative approach	++	++	+++	+++
Tooth sensitivity reduction	++	++	+++	+++
Tooth preparation	+++	+++	+	++
Restoration strength (shear & bond)	++	+	+++	+++
Occlusal contact stability	++	+	+++	+++
Interproximal contact stability	++	+	+++	+++
Pulpal trauma	+	++	+++	+++
Esthetics	++	++	-	+++
Technique sensitivity	++	++	+++	++
Cost	++	++	++	+
Predictability	+	+	+++	+++
Definitive restoration	+++	+++	-	+++
Restoration longevity	++	+	+++	+++

+ Poor; ++ Fair; +++ Excellent.

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
