# Peer review of "An Innovative Treatment Approach Using Digital Workflow and CAD-CAM Part 2: The Restoration of Molar Incisor Hypomineralization in Children"

_ijerph, 2020, doi:10.3390/ijerph17051499_

Round 1
Reviewer 1 Report
must present a final image of the case, without a rubber dam
Author Response
Point 1: must present a final image of the case, without a rubber dam
Response 1: At the end of the treatment, the patient was without sedation, cooperation was poor, and she did not agree to have any more pictures taken.

Reviewer 2 Report
First of all I would like to compliment the quality of the writing. The comparison between the options of treatment for the child best benefits and his family is noteworthy. I merely regret the lack of patient behavior description towards the use of intra oral scanner as the clinical case includes an oral sedation which in a way hides the cooperation of the patient.
I also wished the tooth preparation protocol to be more described (removal of porosities combined with shape preparation for a milled restauration Vs. non milled, immediate dentin sealing considering the high sensitivity of MIH teeth waiting for final restoration despite temporary material)
The paper anyway opens perspective studies in CADCAM restaurations for children and moreover using chairside procedures for faster treatments with regards to patient lack of cooperation.
Thank you very much.
Author Response
Point 1: First of all I would like to compliment the quality of the writing.
The comparison between the options of treatment for the child best benefits and his family is noteworthy.
Response 1: Thank you for the positive comment.
Point 2: I merely regret the lack of patient behavior description towards the use of intra oral scanner as the clinical case includes an oral sedation which in a way hides the cooperation of the patient.
Response 2: A paragraph about the child’s behaviour was added.
Point 3: I also wished the tooth preparation protocol to be more described (removal of porosities combined with shape preparation for a milled restauration Vs. non milled, immediate dentin sealing considering the high sensitivity of MIH teeth waiting for final restoration despite temporary material)
Response 3 -This was added to the text, thank you
Point 4: The paper anyway opens perspective studies in CADCAM restaurations for children and moreover using chairside procedures for faster treatments with regards to patient lack of cooperation.
Response 4: We appreciate the comment.

This manuscript is a resubmission of an earlier submission. The following is a list of the peer review reports and author responses from that submission.